# Replacement Process of Carbonate Apatite by Alveolar Bone in a Rat Extraction Socket

**DOI:** 10.3390/ma14164457

**Published:** 2021-08-09

**Authors:** Xiaoxu Zhang, Ikiru Atsuta, Ikue Narimatsu, Nobuyuki Ueda, Ryosuke Takahashi, Yuki Egashira, Jing-Qi Zhang, Jiong-Yan Gu, Kiyoshi Koyano, Yasunori Ayukawa

**Affiliations:** 1Section of Implant and Rehabilitative Dentistry, Division of Oral Rehabilitation, Faculty of Dental Science, Kyushu University, Fukuoka 812-8582, Japan; xiaoxu3535@dent.kyushu-u.ac.jp (X.Z.); narimatu.i@dent.kyushu-u.ac.jp (I.N.); nobuyuki@dent.kyushu-u.ac.jp (N.U.); tkhsrsk@dent.kyushu-u.ac.jp (R.T.); egashirayuki@dent.kyushu-u.ac.jp (Y.E.); ayukawa@dent.kyushu-u.ac.jp (Y.A.); 2Division of Advanced Dental Devices and Therapeutics, Faculty of Dental Science, Kyushu University, Fukuoka 812-8582, Japan; koyano@dent.kyushu-u.ac.jp; 3Department of Molecular Cell Biology and Oral Anatomy, Faculty of Dental Science, Kyushu University, Fukuoka 812-8582, Japan; zhangjq@dent.kyushu-u.ac.jp (J.-Q.Z.); gujy@dent.kyushu-u.ac.jp (J.-Y.G.)

**Keywords:** carbonate apatite, bone substitute, autogenous bone, osteoclast

## Abstract

The objective of this study was to investigate a bone graft substitute containing carbonate apatite (CO_3_Ap) to analyze bone replacement and the state of bone formation in vitro and in vivo compared with autogenous bone (AB) or control. An osteoclast precursor cell line was cultured with AB or CO_3_Ap, and morphological analysis using scanning electron microscopy and a tartrate-resistant acid phosphatase activity assay were performed. The right maxillary first and second molars of Wistar rats were extracted and compensated by AB or CO_3_Ap granules. Following implantation, the bone formation state was evaluated after 3, 5, 7, 14, and 28 days of surgery by micro-computed tomography and immunohistostaining. The osteoclast-like cell morphology was typical with many cell protrusions in the AB and CO_3_Ap groups. Additionally, the number of osteoclast-like cells formed in the culture increased in each group; however, there was no significant difference between the AB and CO_3_Ap groups. Five days after tooth extraction, osteoclasts were observed near CO_3_Ap. The bone thickness in the CO_3_Ap group was significantly increased than that in the control group and the bone formation in the CO_3_Ap group increased by the same level as that in the AB group. CO_3_Ap is gradually absorbed by osteoclasts in the extraction socket and is easily replaced by alveolar bone. The process of bone replacement by osteoclasts is similar to that of autologous bone. By observing the process of bone replacement in more detail, it may be possible to gain a better understanding of the bone formation and control the amount of bone after surgery.

## 1. Introduction

Alveolar bone resorption is an important issue in all types of defective prosthesis treatment, including dental implants, and osteogenesis is necessary to repair the bone. There are various bone grafting materials available for bone augmentation; however, clinical application requires an understanding of their various advantages and disadvantages. Of these, autogenous bone is the first choice for bone augmentation materials [1].

Autogenous bone is the gold standard bone replacement material because of its superior osteoconductive and osteoinductive properties [1,2]. Osteogenesis requires a bone-replacement material as a scaffold for healing, in which the scaffold does not damage or introduce antigenicity to the surrounding tissue, promotes vascularization and osteogenesis, and can easily be replaced with new bone after filling. Only fresh autogenous bone can successfully and effectively replace bone. Intuitively, one can think of the null hypothesis as stating that any artificial bones are not as good as autogenous bones.

However, autogenous bone grafting has two major disadvantages. First, in elderly patients and patients with large bone defects, the collection of autogenous bone is a serious surgery and the amount of bone collected is limited [2]. Second, it is difficult to control the amount of bone after autogenous bone grafting [3] because even if similar or the same quality bone is added to the site where bone resorption originally occurred, it can easily be resorbed again by remodeling. Contrary to autogenous bone, artificial bone is not similarly absorbed by the body to enable space for osteoblast activity until sufficient bone is repaired [4].

A typical material for artificial bone is hydroxyapatite (HA), which is a tooth enamel component. However, because HA remains at the site without being replaced for a long period of time, there is a risk of infection [5,6]. It is important to have a composition similar to that of bone for biocompatibility; however, when considering “space making” for bone formation, it is also necessary to have a low absorption rate. Therefore, an artificial material with a composition similar to that of autogenous bone, such as a bone substitute containing carbonate apatite (CO_3_Ap) that is an inorganic component of bone, has gained clinical attention [7,8]. Such composites have been clinically applied for excellent results; however, it is difficult to predict the control of absorption clinically, which is critical to bone regeneration [8]. We previously focused on using osteoclast activity as an evaluation method of bone remodeling. Osteoclasts also initiate bone formation [9].

In this study, we investigate the process in which CO_3_Ap in an extraction socket is absorbed in vivo and replaced with bone. We considered the mechanism in which CO_3_Ap used for bone augmentation in clinical practice is replaced with bone.

## 2. Materials and Methods

### 2.1. Materials

Autogenous bone (AB) and CO_3_Ap were used in the experiment as bone filling materials. AB was collected from the femur and tibia of three male 6-week-old Wistar rats with a bone scraper (Osteogenics, Lubbock, TX, USA). Cytrans^®^ (GC, Tokyo, Japan) is a pure CO_3_Ap dense granule fabricated in an aqueous solution through a dissolution precipitation reaction using Ca(OH)_2_ granules as precursors [10,11,12,13]. The CO_3_Ap granules were 300–600 μm in diameter.

### 2.2. Osteoclast Precursor Cell Line RAW-D Cells Culture

Osteoclast-like cells were differentiated from murine osteoclast precursor cell line RAW-D cells (a subclone of RAW264 cells) as previously described [14,15,16]. Briefly, RAW-D cells were cultured in α-modified Eagle’s medium (α-MEM, Wako, Osaka, Japan), 10% fetal bovine serum (FBS, Thermo Fisher Scientific, Waltham, MA, USA) in the presence of recombinant human soluble receptor activator NF-κB ligand (RANKL) (50 ng/mL, Oriental Yeast, Tokyo, Japan) for 4 days (Figure 1A).

### 2.3. Tartrate-Resistant Acid Phosphatase (TRAP) Activity Assay

Osteoclasts were examined using a commercial TRAP staining kit (Sigma-Aldrich, St. Louis, MO, USA) for the osteoclast marker TRAP activity [17]. Following incubation for 4 days, the cells were fixed in 10% formalin (Wako, Osaka, Japan), permeabilized using ethanol/acetone (Wako, Osaka, Japan), and observed with light microscopy (BZ-X800, Keyence, Osaka, Japan). Fifteen samples (five per groups) were analyzed in the TRAP activity assay.

### 2.4. Number of TRAP-Positive Cells

RAW-D cells (6.8 × 10^3^/mL) were seeded on experimental and control groups. AB and CO_3_Ap were ground to a particle size of 200 μm or less and added at a concentration of 50 ng/mL RANKL for 4 days. The number of multinucleated TRAP-positive cells was counted, converted per well, and the average was compared with the control. The samples were observed with light microscopy (BZ-X800, Keyence, Osaka, Japan). The number of TRAP-positive cells was calculated using cell count of BZ-X800 Analyzer software version 1.1.1.8 (Keyence, Osaka, Japan).

### 2.5. Scanning Electron Microscopy

Bovine dentin pieces (8.0 mm × 4 mm, GC, Tokyo, Japan) and CO_3_Ap plates (*φ*9.0 mm × 1 mm provided by GC, Tokyo, Japan) were immersed in α-MEM containing 10% FBS in the presence of 50 ng/mL RANKL at 4 °C for 4 days. RAW-D cells (6.8 × 10^3^/mL) were seeded on the materials. RAW-D cells in culture dish (Nalge Nunc, Rochester, NY, USA) were used as the control. The RAW-D cell morphology on the control culture dish or CO_3_Ap plates, and bovine dentin pieces was evaluated by scanning electron microscopy (SEM) (S-3400N, Hitachi, Tokyo, Japan) at 15 kV. All samples were fixed with 2.5% glutaraldehyde for 30 min. The fixed cells were rinsed in phosphate buffer saline (PBS), dehydrated in a graded ethanol series, and lyophilized. Specimens were coated with Au/Pd alloy and evaluated microscopically [18].

### 2.6. Animals

Fifteen male Wistar rats (6-week-old, body weight 160–180 g, five per groups) were used in the in vivo experiments.

All experiments were performed in accordance with the ARRIVE Guidelines for reporting animal research [19]. All procedures involving experimental animals were approved by the Institutional Animal Care and Use Committee of Kyushu University (Approval Number: A25-240-0).

### 2.7. Tooth Extraction Socket Model

The experimental schedule is shown in Figure 2A. The right maxillary first (M1) and second molars (M2) were extracted under a combination anesthetic (0.3 mg/kg of medetomidine, 4.0 mg/kg of midazolam, and 5.0 mg/kg of butorphanol) [20]. After extraction, the alveoli were filled with AB, CO_3_Ap, or without filling and the animals were randomly selected into three groups (control, AB, and CO_3_Ap).

### 2.8. Micro-Computed Tomography

Each day after the rats were sacrificed, their maxillae were corrected, and fixed in 4% paraformaldehyde (Merck, Darmstadt, Germany) for 24 h. Micro-computed tomography (micro-CT) imaging was performed by micro-CT (SkyScan 1076, Bruker, Kontich, Belgium) using X-ray energy with a tube current of 201 μA and voltage of 49 kV. Three-dimensional analysis software version 1.18.8 (CTAn, Bruker, Kontich, Belgium) was used for analysis [21].

### 2.9. Bone Histomorphometry

An image of the frontal fracture was captured using micro-CT, and the vertical length to the base of the extraction socket was measured as the bone thickness. The anteroposterior position was located by the central part of the second molar (maximum crown diameter) on the opposite side.

### 2.10. Tissue Preparation and Histological Staining

Tissues were prepared as described in a previous study [18,22]. Briefly, at each time point, the rats were sacrificed and their maxillae were isolated and immersed in 4% paraformaldehyde for 24 h, followed by decalcified in Kalkitox^TM^ solution (Wako, Osaka, Japan) exposure at 4 °C for 18 h. The frozen samples were immersed in 20% sucrose overnight at 4 °C. The samples were next embedded in Optimal Cutting Temperature compound (Sakura Finetek, Tokyo, Japan) at 4 °C for 2 h and then cut into 10-μm-thick sagittal sections using a cryostat (CM1860, Leica Microsystems, Wetzlar, Germany) at −20 °C. Sections were stained with hematoxylin and eosin (HE) for recognizing tissue types and the morphologic changes or Azan stains for the collagen fibers (Figure 3A,B) [23]. In addition, some sections were subject to TRAP staining for the osteoclasts [17].

### 2.11. Immunofluorescence Staining Procedure

Sections prepared from the CO_3_Ap group 5 days after tooth extraction were immunostained. After unreacted aldehyde groups were quenched with 10 mM glycine, sections were blocked in 10% chick albumin for 60 min at room temperature, followed by further blocking in 10% normal goat serum for 2 h at room temperature. Sections were then incubated with a mixture of Acti-stain^TM^ 488 Fluorescent Phalloidin (1:40 dilution, Cytoskeleton, Denver, CO, USA) to detect F-actin filaments, and anti-cathepsin K antibody (1:50 dilution, Santa Cruz Biotechnology, Santa Cruz, CA, USA) to detect osteoclast or control IgG at room temperature for 4h in a moisture chamber. The sections were washed with PBS and incubated with goat anti-mouse IgG (H + L) antibody conjugated with Alexa Fluor 568 (1:300 dilution, Invitrogen, Waltham, MA, USA) for 30 min [24,25]. Nuclei were stained with 4′,6-diamidino-2-phenylindole (DAPI) (1:50 dilution, Sigma-Aldrich, St. Louis, MO, USA). The samples were observed with Apotome 2 microscopy (Zeiss, Oberkochen, Germany).

### 2.12. Statistical Analysis

Our experiment used 5 samples and all data were expressed as means ± standard deviation (SD) from at least three independent experiments. One-way analysis of variance (ANOVA) with Tukey’s test was performed. Values of *p* < 0.05 were considered significant.

## 3. Results

### 3.1. Number of Osteoclasts

The in vitro culture experiment was performed according to the time schedule shown in Figure 1A. As shown in Figure 1B, the number of TRAP-positive cells increased significantly on AB and CO_3_Ap compared with the control. However, there were no significant differences between AB and CO_3_Ap groups.

### 3.2. Morphological Changes in the Osteoclasts

As shown in the SEM image in Figure 1C, many oval osteoclast-like cells were observed in the control group, and stellate and extended osteoclast-like cells were observed in the AB and CO_3_Ap groups. In addition, cell protrusion tips adhered to the material surface.

### 3.3. Chronological Change of the Bone Amount

Animal experiments were performed according to the time schedule shown in Figure 2A. The extraction socket was observed by a frontal cut using micro-CT (Figure 2C) with the central part of the second molar on the opposite side a measurement landmark (Figure 2B, red dotted line). Compared with the surrounding bone, CO_3_Ap shows a high CT value and appears as high density on the CT images (Figure 2D). Surfaces with CO_3_Ap in the extraction socket were observed until day 28, and it was confirmed that the sites were replaced with new bone on day 14. And the granule size was reduced day-by-day. The thickness of the bone was also evaluated using a micro-CT image (Figure 2E). Five days after tooth extraction, a significant increase in the thickness of the bone in AB and CO_3_Ap groups was observed compared with the control group. Twenty-eight days after tooth extraction, the thickness of the new bone significant increase was only observed in the CO_3_Ap group compared with the control group.

### 3.4. Chronological Change of the Alveolar Bone Morphology after Tooth Extraction

The extraction socket in each group was stained with HE and observed over time to understand the healing process (Figure 3A). Accumulation of inflammation-related cells, such as neutrophils, was not observed in all groups, and inflammatory findings, such as infectious granulation, were also not observed. Five days after tooth extraction, a rapid recovery of bone thickness was observed in the AB and CO_3_Ap groups. Seven days after the extraction, the bone thickness around the extraction socket of the AB and CO_3_Ap groups was flatter than the surrounding alveolar bone thickness (red dots line in Figure 3B). In addition, angiogenesis was observed around the CO_3_Ap granules (black arrowheads in Figure 3A).

As shown in Figure 3B, Azan staining was performed using a serial section compared with the HE staining in Figure 3A. Immature bone formation was observed in the AB and CO_3_Ap group 5 days after tooth extraction, and rapid bone formation was observed in the AB group and CO_3_Ap group 14 days later. Twenty-eight days after tooth extraction, densification of new bone was observed in the AB group and CO_3_Ap group.

### 3.5. Activity of Osteoclasts around CO_3_AP

Five days after tooth extraction, TRAP staining revealed osteoclasts accumulating around CO_3_Ap (Figure 4A). Large cells with multiple nuclei were observed (black arrowheads in Figure 4A). Furthermore, cathepsin K, another marker of osteoclast, was used to stain serial sections, and positive cells were observed around CO_3_Ap (white arrowheads in Figure 4B).

## 4. Discussion

In this study, the process of CO_3_Ap replacement to bone was observed over time using a rat molar extraction socket model. The method for observing wound healing of the extraction socket has been widely used for evaluating bone formation [26]. Alternatively, there have been many previous studies evaluating bone formation using femur and calvaria models [7,27]. However, because the maxilla has more blood flow than other bone and the trabecular structure is a special environment, where various cells can move in the bloodstream at wide intervals, the results of this study are considered to be directly linked to clinical practice. Therefore, the extraction socket model was selected.

In the healing of human extraction sockets, the period of up to approximately 1 week is called the “clot stage”, during which fibrin has a hemostatic effect and connective tissue proliferation occurs. The “granulation tissue stage” is at approximately 2 weeks, and as the epithelium heals, macrophages organize the blood clot. Around 3–4 weeks is the “callus stage”, which is the time when granulation is replaced with new bone. After 5 weeks, the “bone remodeling stage” occurs and the bone tissue is strengthened by bone remodeling. The starting point of healing in this series is the presence of osteoclast progenitor cells, such as macrophages. Bone remodeling usually starts when osteoclasts appear, the signal coupling from osteoclasts promotes osteoblast bone formation and leads to neoplastic bone formation. Therefore, in this study, we focused on the activity of osteoclasts in the extraction socket as a method for evaluating bone formation.

In the experiment shown in Figure 1, RAW-D cell, an osteoclast precursor cell line, was used. This cell line was isolated from RAW264 cells in 2004 and has since been used in many basic studies to evaluate osteoclast behavior [15,28,29]. In these experiments, which showed the ability to differentiate into osteoclast-like cells in the AB and CO_3_Ap experimental groups, the number of multinucleated osteoclast-like cells increased comparably to AB in the presence of CO_3_Ap (Figure 1B). Furthermore, the thick protrusion shape and bearing three or more nuclei of the osteoclasts on the CO_3_Ap was a characteristic result. In other words, the control group supported osteoclast-like cells with an oval shape, and the cells on the experimental group were star-shaped and large (Figure 1C). Given the number and shape of the cells indicate the strength of the activity of osteoclasts [30], the osteoclast-like cells on the CO_3_Ap and autogenous bone showed high TRAP activity. Previous studies have reported that the rougher the implant body surface, the greater the number of multinuclear giant cells that appear. Therefore, the results in this experiment may be influenced by the composition and the surface roughness of the materials. Alternatively, CO_3_Ap releases large amounts of calcium and phosphate ions into solution [30,31,32], which may affect the activity of osteoclasts [30].

The process of bone replacement in the CO_3_Ap group over time was observed using the rat tooth extraction socket model. The observation period shown in Figure 2 was determined according to a previous study [33,34]. In a previous experiment in which CO_3_Ap was implanted in a rabbit femur, bone replacement was observed over 24 months using micro-CT images. However, it is difficult to compare the time of bone replacement because it is affected by the healing environment and wound size that can occur with different models. In rabbit and rat tibia fracture models, the rate of healing slows with body size [35,36,37]. The sagittal section through the central part of the second molar on the opposite side is shown in Figure 2C,D. For the bone thickness measurements shown in Figure 2E, the palatal root apex of the micro-CT image in Figure 2D was used as a landmark from which the vertical bone thickness was measured. The bone was significantly thicker in both the AB group and the CO_3_Ap group on day 5 in Figure 2E. However, because of the small size of the extraction socket in the present model, the experimental group of the bone replacement material caught up with the control group after approximately 14 days. If large animals, such as humans and dogs, were used as models, significant differences might be obtained. After 28 days of tooth extraction, only the CO_3_Ap group showed a predominantly thicker bone than the control group. Over the long term, there may be a difference between CO_3_Ap and AB group.

The morphology of the same site was observed under HE staining (Figure 3A). An inflammatory reaction was not observed immediately after the bone substitute was transferred. Defective granulation tissue was not observed, while angiogenesis was clearly enhanced around CO_3_Ap. Bone filling materials are foreign substances and may cause inflammation depending on their shape and size [32,35]. In addition, once infection occurs, the granulation tissue, which should be the starting point for healing, becomes poor granulation tissue and conversely suppresses healing [38]. It is considered that the soft tissue barrier is formed early by being filled with the bone filling materials [39,40]. In addition, because vascular endothelial cells can be easily determined from the HE-stained image, the enhancement of angiogenesis could be confirmed. Based on these results, osteoclast accumulation appeared around CO_3_Ap, which may lead to early bone renewal. Furthermore, from the results of Azan staining in Figure 3B, collagen fibers (type I collagen, a main component of the organic part of bone) were abundant according to the staining concentration. The presence of collagen fibers suggests that the stained area is still immature [41]. Therefore, there was immature bone formation in the experimental group with strong staining of collagen fibers after 5 days. Collagen fiber was also observed in the control group, although not as prominently as in the experimental groups.

The osteoclast activity was observed (Figure 4) because osteoclasts attach to juvenile bone to initiate bone formation [42]. Cells with multiple nuclei and clearly larger than other cells were observed around CO_3_Ap (Figure 4A). The cells with these characteristics are likely to be osteoclasts.

Furthermore, cathepsin K-positive cells were observed around CO_3_Ap (Figure 4B). Because cathepsin K is a marker of osteoclast [24], the osteoclasts were activated around CO_3_Ap and consequently bone formation was promoted. In addition, because osteoclasts were observed around the bone filling material and throughout the extraction socket with TRAP staining, CO_3_Ap absorption occurs simultaneously in the entire sample. This is clearly different from the bone replacement of other materials that occur from the margins [43,44], and the material may be quickly replaced by bone.

In aqueous solution, CO_3_Ap releases ions, such as carbonic acid, calcium, and phosphoric acid, which easily reduce the pH to create a suitable environment for osteoclasts. Furthermore, CO_3_Ap is stable at in vivo pH and dissolves well at extremely low pH [31]. Therefore, the reduced pH by osteoclasts elutes carbonate apatite and releases ions, leading to a cycle in which the pH is further reduced. Furthermore, as shown in Figure 1C, CO_3_Ap is an environment in which osteoclast-like cells are likely to adhere.

Autogenous bone is an excellent material as a bone substitute; however, the amount of absorption is easily affected by the limitations of the filling amount and surrounding bone environment. Furthermore, the hardening is easily affected by the collection site and it is difficult to control the prognosis of treatment. Understanding the process by which CO_3_Ap, an artificial material used as a bone substitute, is indispensable for controlling the prognosis. Therefore, CO_3_Ap is a bone substitute that potentially meets clinical needs more than autogenous bone.

## 5. Conclusions

In this study, the bone replacement process of CO_3_Ap in the extraction socket model was observed. Although CO_3_Ap is a synthetic material, the process of bone replacement by osteoclasts was similar to that of autogenous bone. By observing the process of bone replacement in more detail in the future, it may be possible to obtain a better understanding of the bone formation compared with autogenous bone.

## Figures and Tables

**Figure 1 materials-14-04457-f001:**
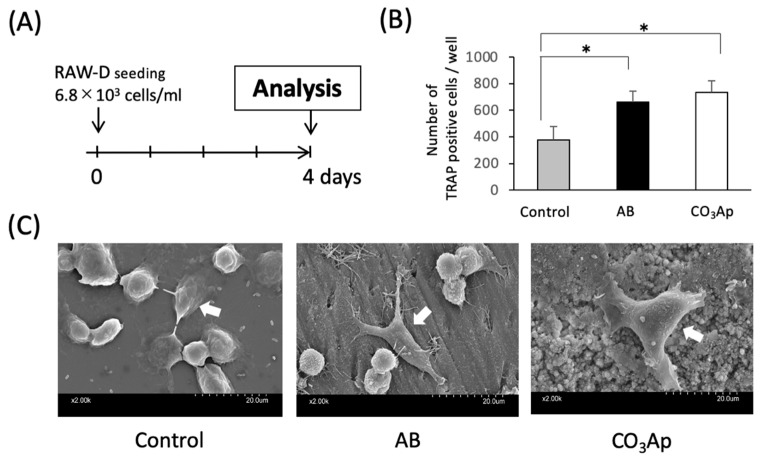
**Evaluation of osteoclast activity in an in vitro study.** (**A**) Experimental protocol of the in vitro study. (**B**) The number of TRAP positive cells (*n* = 5, * *p* < 0.05). (**C**) The shape of the osteoclast-like cells (white arrows) (control: culture dish, AB: dentin piece, CO_3_Ap: Cytrans disc).

**Figure 2 materials-14-04457-f002:**
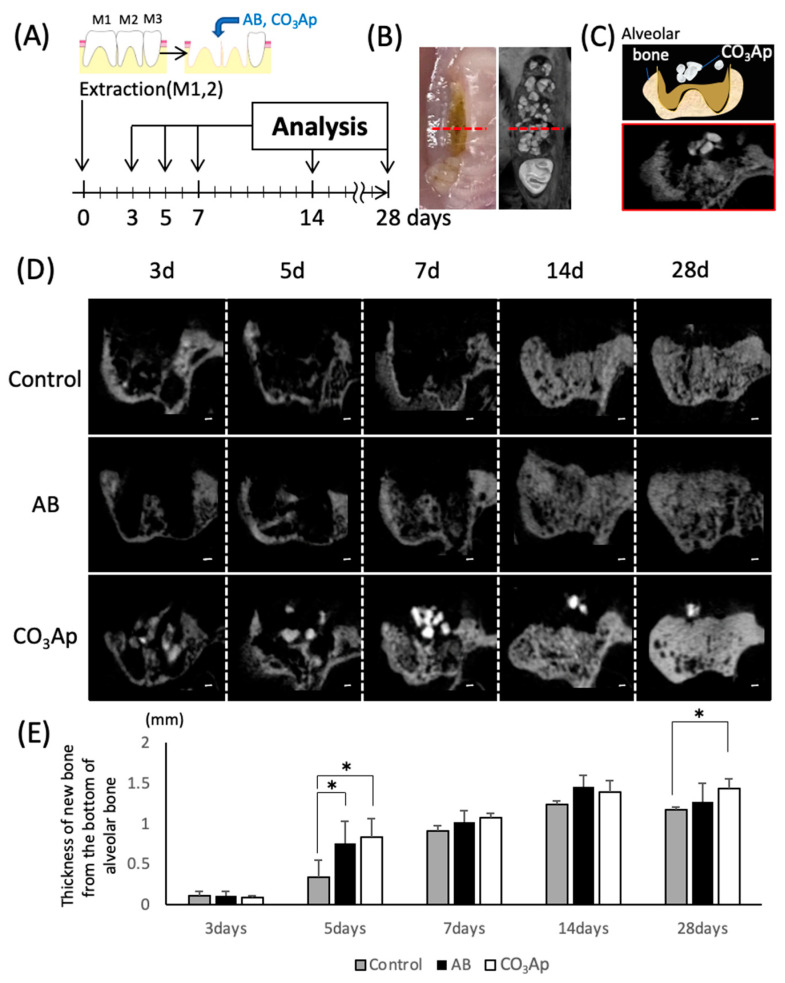
**Evaluation of the chronological bone formation with micro-CT.** (**A**) Schematic of the tooth extraction model. The right maxillary first (M1) and second molars (M2) were replaced by bone substitute. Experimental protocol of the in vitro study as described in the Materials and Methods section. (**B**) Landmark of the micro-CT image. The anteroposterior position at that time was located by the central part of the second molar (maximum crown diameter) on the opposite side. (**C**) Image of the frontal section at the observation site (schematic and CT image). (**D**) Evaluation of chronological changes in the extraction socket using micro-CT (control: extraction socket, AB: with autogenous bone, CO_3_Ap: with Cytrans). Bars = 200 μm. (**E**) Thickness of the new bone from the bottom of the alveolar bone (*n* = 5, * *p* < 0.05).

**Figure 3 materials-14-04457-f003:**
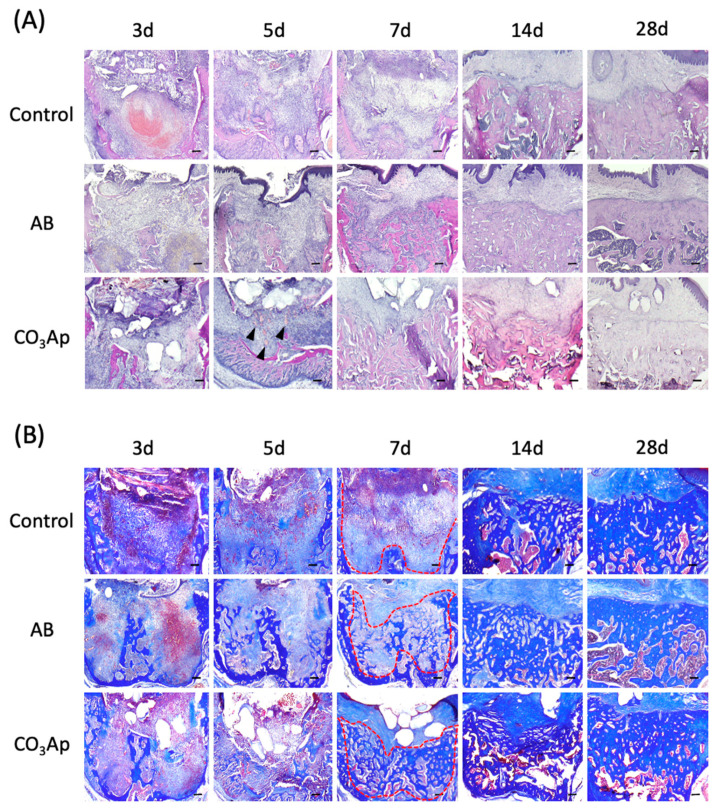
**Chronological change of the alveolar bone morphology after tooth extraction.** (**A**) Evaluation using HE staining. The black arrowheads indicate angiogenesis. Bars = 100 μm. (**B**) Azan staining shows collagen-rich new bone and the red dots line indicates border of the bone. Each slice shows the serial section from (**A**). Bars = 100 μm.

**Figure 4 materials-14-04457-f004:**
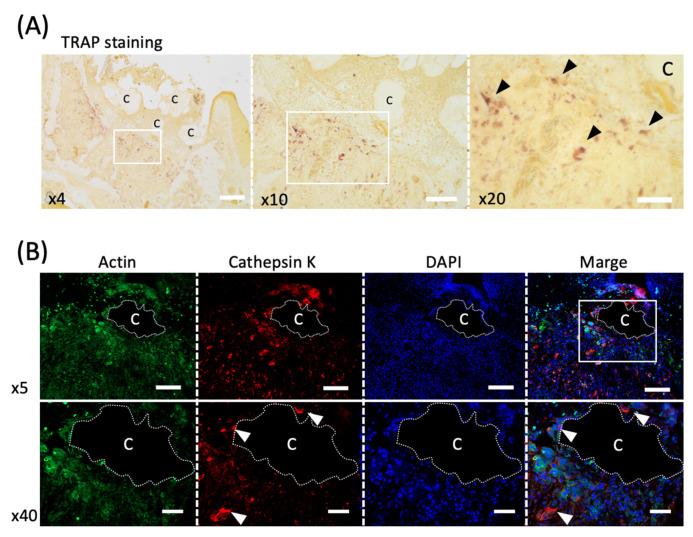
**Evaluation of the osteoclast activity in an in vivo study.** (**A**) TRAP staining of the healing socket 5 days after tooth extraction. The black arrowheads indicate osteoclasts. C; Cytrans. Bars = 300 μm (×4), 200 μm (×10), 100 μm (×20). (**B**) Immunofluorescence staining of cathepsin K (red), F-actin (green) and DAPI (nuclei, blue) in upper panel (×5) and lower panel (×40). White arrowheads indicate multinucleated osteoclasts co-localized with cathepsin K and DAPI. C; Cytrans (white dots area). Bars = 500 μm (×5), 100 μm (×40).

## Data Availability

The data presented in this study are available on request from the corresponding author.

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
