# Peer review of "Replacement Process of Carbonate Apatite by Alveolar Bone in a Rat Extraction Socket"

_materials, 2021, doi:10.3390/ma14164457_

Round 1

Reviewer 1 Report

The paper reports different in vitro and in vivo animal study on the biological integration of graft materials including autogenous bone and carbonate apatite granules.The topic is interesting, and the article is generally well written. However a number of changes and improvements must be done.

Abstract

- Control group should be mentioned in the materials and methods subheading.

Introduction

Line 68-69 The last sentence can be omitted.

Materials and methods

- Have you followed ARRIVE guidelines for animal model studies (https://arriveguidelines.org/)?

- How many samples per groups were analyzed in the in vitro study? How many independent experiments were conducted?

- The methods for SEM analysis of the in vitro study present some inconsistency.  In the paragraph 2.5 “bovine teeth” are mentioned as the substrate for osteoclast culture. In figure 1C caption it is stated that “autogenous bone from cranium” was used. Please check and clearly explain the exact materials that have been used. What is the rationale for using bovine teeth to resemble autogenous bone?

- Paragraph 2.6. The grafting procedure should be described in more detail. How much graft material was used? Was the post-extractive alveolus completely filled by the material? What kind of healing did occur (secondary/primary?) Had barrier membranes been used? What was the procedure for the control group (blood coagulum/collagen?)

Results

- Line 176. What does “white sites” mean?

- Line 178. How can you measure granular size by micro-CT? Do you think a dimensional reduction can be observed after only 14 days of osteoclast activity?

- Figure 3. Looking at the micrograph and the granules seem to have reduce their number rather than in dimensions. Some granules seem to be embedded in soft tissue. Please discuss these observations.

Discussion

- Line 275-276: many limitations affect the strength of the results regarding the effectiveness in ridge preservation, mainly the small number of samples and the animal model characteristics. Please re-elaborate the sentence.

- Lines 279-283. Please discuss the importance of artificial or soft tissue barrier formation to protect graft materials from infection (see the works by Laurito et al PMID: 27740650; DOI: 10.11607/prd.2114 and     PMID: 28196172 DOI: 10.11607/prd.2731)

- Line 303: Please avoid rhetorical question.

- Study limitation must be enumerated and discussed.

Author Response

Abstract:

  1. Control group should be mentioned in the materials and methods subheading.

> Response: Thank you for your comments. We have added the following sentences to the revised manuscript (page 2-3).

“…Fifteen samples (five per groups) were analyzed in the TRAP activity assay.”

“…After extraction, the alveoli were filled with AB, CO3Ap or without filling and…”

Introduction:

  1. Line 68-69 The last sentence can be omitted.

> Response: Thank you for the advice. We removed the sentence.

Materials and Methods:

  1. Have you followed ARRIVE guidelines for animal model studies (https://arriveguidelines.org/)?

> Response: Yes, we have. We have added the sentences about it. (page 3).

  1. How many samples per groups were analyzed in the in vitro study? How many independent experiments were conducted?

> Response: Thank you for your question. Our experiment used 5 samples in each group and a priori Shapiro–Wilk test was performed to test for normality. And we added the following to the revised manuscript (page 2-3). Please check them.

“…Fifteen samples (five per groups) were analyzed in the TRAP activity assay.”

“…All deta were expressed as means ± standard deviation (SD) from at least three independent experiments.”

  1. The methods for SEM analysis of the in vitro study present some inconsistency.  In the paragraph 2.5 “bovine teeth” are mentioned as the substrate for osteoclast culture. In figure 1C caption it is stated that “autogenous bone from cranium” was used. Please check and clearly explain the exact materials that have been used. What is the rationale for using bovine teeth to resemble autogenous bone?

> Response: In our previous in vitro experiments with osteoclasts, “Ivory” or “bovine teeth” have been used in place of autologous bone (Ogawa et al., J Bone Miner Metab 2007). However, ivory has recently been in ethical issue, so in this study we experimented with bovine teeth only in vitro.

  1. Paragraph 2.6. The grafting procedure should be described in more detail. How much graft material was used? Was the post-extractive alveolus completely filled by the material? What kind of healing did occur (secondary/primary?) Had barrier membranes been used? What was the procedure for the control group (blood coagulum/collagen?)

> Response: In this study, a rat tooth extraction socket was used, and we fixed the quantify by filling the socket with materials. In addition, as the reviewer said, we tried to use a collagen membrane or the connective-tissue transplanting in our previous experiment methods, but there was not much difference in these models. On the contrary, membranes and grafts sometimes interfered with healing. It is considered that this is because the size of the wound site was small in the rats used, so it was difficult to perform operations such as placing the membrane, and the healing speed was usually fast. That is why the advantages of the membrane could not be utilized.

Therefore, in this study, the socket was not covered with anything, and only with the clot at the time of hemostasis.

Results:

  1. Line 176. What does “white sites” mean?

> Response: Thank you for pointing out. This meant CO3Ap, but it was difficult to understand, so we revised the text.

  1. Line 178. How can you measure granular size by micro-CT? Do you think a dimensional reduction can be observed after only 14 days of osteoclast activity?

> Response: Thank you for pointing out. In this study, we measured the height of new bone edge, not the size of CO3Ap granular. Therefore, it was only a subjective opinion that the granular became clearly smaller not only after the 14th day but also over time. The text was partially revised because it was difficult to understand.

  1. Figure 3. Looking at the micrograph and the granules seem to have reduce their number rather than in dimensions. Some granules seem to be embedded in soft tissue. Please discuss these observations.

> Response: Indeed, the presence of large amounts of CO3Ap into soft tissues can block the soft-tissue healing.

However, when the amount of material was small, the result was not so affected to the presence of materials. And no inflammatory findings were observed. This was described in "Result". Furthermore, the influence of CO3Ap with soft tissues is being submitted as a new paper.

Discussion:

  1. Line 275-276: many limitations affect the strength of the results regarding the effectiveness in ridge preservation, mainly the small number of samples and the animal model characteristics. Please re-elaborate the sentence.

> Response: Thank you for your comment. And we agree with you, so we deleted the following sentence. “…, so the effectiveness of CO3Ap was sufficiently demonstrated.”

  1. Lines 279-283. Please discuss the importance of artificial or soft tissue barrier formation to protect graft materials from infection (see the works by Laurito et al PMID: 27740650; DOI: 10.11607/prd.2114 and     PMID: 28196172 DOI: 10.11607/prd.2731)

> Response:  Thank you for your comments. We added the sentence and two references to the text. (page 8).

“…It is considered that the soft tissue barrier is formed early by being filled with the bone filling materials.”

  1. Line 303: Please avoid rhetorical question.

> Response: We are sorry. It is our mistake… We have delated the sentence “So why did bone form around CO3Ap?”.

  1. Study limitation must be enumerated and discussed.

> Response: Thank you for your important point. Added to the 4th sentence of “Discussion”.

Reviewer 2 Report

ABSTRACT

“…In this study, we investigate a bone graft substitute containing…” – the authors must replace this sentence with: “the objective of this study is…”

Line 19 - “…Maxillary first and second molars of male 6-week-old Wistar rats were extracted…”

From which side?Right side or left side?

Line 20 - “Following transplantation…”

The authors wanted to say: following implantation?

INTRODUCTION

Line 40 – the reference is missing.

“…In this study, we investigate a bone graft substitute containing…” – the authors must replace this sentence with: “the objective of this study is…”

The null hypothesis is missing.

MATERIALS AND METHODS

Line 75 - “…AB was collected from the femur and tibia of male 6-week-old Wistar rats…”

Why were rats used so young? How many animals were used for bone harvesting? Why was bone harvested from long bones and not from bone with the same embryological origin as the place where the grafts were then implanted?

Line 114 - “…The maxillary first (M1) and second molars (M2) were extracted…”

How many rats were used? What was the weight of the animals? Did the authors use the same animals from which they harvested the bone? The extracted teeth were from the right or the left side? How many teeth were extracted? How was the extraction performed? They used any instrument adapted for this purpose?

Line 116 - “…After extraction, the animals randomly categorized into three groups (control, AB, CO3Ap)…”

authors must replace this sentence with:

After extraction, the alveoli were filled with AB, CO3Ap or without filling and the animals randomly categorized into three groups…”

How was the randomization of the animals performed? After implantation of materials, were the alveoli sutured? If yes, which suture was used. If not, why? How was the implanted material stabilized?

Line 120 - “ …After the rats were sacrificed…”

How long after implantation were they sacrificed?

How many animals were used for microCT and how many for histology?

Line 137- “…The samples were next embedded in O.C.T. compound (Sakura Finetek…”

What is O.C.T.?

Line 144 - “ …Sections prepared from the CO3Ap group 5 days after tooth extraction…”

And the samples with AB and with the empty alveoli were not analyzed?

RESULTS

Fig. 1.AB: autogenous bone from cranium”

On the materials, is it not mentioned that AB from cranium was used?

On Fig 2A the analysis was performed at 3, 5 and 14 days, but in the figure 2D and E the authors presents results at 3, 5, 7, 14 and 28 days. What should we accept? On materials and methods this is not mentioned!

DISCUSSION

Line 234 - “…In the healing of human extraction sockets, the period up…”

What is the comparison with the rat? Why were the periods of 3, 5, 7,…days used?

Line 229 - ” … many previous studies evaluating bone formation using tibia and skull models [26].”

The reference 26 does not support the sentence. The author should refer articles on studies in tibia and cranium. Like for example: Ceramics International 2018; 44 (5): 5025-5031.DOI: 10.1016/j.ceramint.2017.12.099,

CONCLUSIONS

The authors must replace “AB” with autogenous bone

Line 319: “ …Although CO3Ap is an artificial bone…”

This is a synthetic material, not artificial bone. Correct this sentence.

REFERENCES

The authors present very old articles that should be replaced by more recent references on the same subjects

The authors should refer:

: Ceramics International 2018; 44 (5): 5025-5031.DOI: 10.1016/j.ceramint.2017.12.099

Author Response

[Abstract]

  1. “…In this study, we investigate a bone graft substitute containing…”– the authors must replace this sentence with: “the objective of this study is…”

> Response: Thank you for precious advises. We have modified the sentence as your mention.

  1. Line 19 - “…Maxillary first and second molars of male 6-week-old Wistar rats were extracted…”From which side? Right side or left side?

> Response: Thank you for comments. We added the details of Animal models. Please check at “Materials and Methods”.

  1. Line 20 - “Following transplantation…” The authors wanted to say: following implantation?

> Response: Thank you for pointing out. Surely, it means implantation. We corrected the word.

[Introduction]

  1. Line 40 – the reference is missing.

> Response: Thank you for your pointing out. We have added the reference.

  1. “…In this study, we investigate a bone graft substitute containing…”– the authors must replace this sentence with: “the objective of this study is…”

> Response: Thank you for your precious advises. We have modified the sentence.

  1. The null hypothesis is missing.

> Response: We added the following to the revised manuscript (page 1).“…Intuitively, one can think of the null hypothesis as stating that any artificial bones are not as good as autogenous bones.”

[Materials and Methods]

  1. Line 75 - “…AB was collected from the femur and tibia of male 6-week-old Wistar rats…”

Why were rats used so young? How many animals were used for bone harvesting? Why was bone harvested from long bones and not from bone with the same embryological origin as the place where the grafts were then implanted?

> Response: Thank you for your question. Certainly, there are many studies using old rats, but in this study, the age of the collected rats was adjusted to the age of the model rats used in the experiment. The age of the model rats in the experimental group was determined based on past studies in consideration of the ease of tooth extraction (Narimatsu et al., ACS Biomater Sci Eng. 2019; Atsuta et al., J Biomed Mater Res A. 2019; Atsuta et al., Clin Implant Dent Relat Res. 2013).

  1. Line 114 - “…The maxillary first (M1) and second molars (M2) were extracted…”. How many rats were used? What was the weight of the animals? Did the authors use the same animals from which they harvested the bone? The extracted teeth were from the right or the left side? How many teeth were extracted? How was the extraction performed? They used any instrument adapted for this purpose?

> Response: We added the details of Animals and extraction socket models. Please check at Materials and Methods.

“2.6. Animals Fifteen male Wistar rats (6-week-old, body weight 160–180g, five per groups) were used in the in vivo experiments.”

“…AB was collected from the femur and tibia of three male 6-week-old Wistar rats with a bone scraper”

“…The right maxillary first (M1) and second molars (M2) were extracted under a combination anesthetic…”

> Moreover, we use the experimental Extraction forceps which was made by myself before about 10 years.

  1. Line 116 -“…After extraction, the animals randomly categorized into three groups (control, AB, CO3Ap)…”

authors must replace this sentence with: After extraction, the alveoli were filled with AB, CO3Ap or without filling and the animals randomly categorized into three groups…”

> Response: Thank you for your advice. We have modified the sentence as your comment.

  1. How was the randomization of the animals performed? After implantation of materials, were the alveoli sutured? If yes, which suture was used. If not, why? How was the implanted material stabilized?

> Response: Thank you for your question. We just have randomly selected (not categorized) for each experiment. And also, we have changed the sentence in the manuscript.

> We did not suture the mucosa on the extraction socket in this experiment. As the experimental method, we used “sutures”, “collagen membranes”, and “connective tissue transplant” to cover the socket, but there was not much difference in this model. It is considered that this is because the size of the wound site was small in the rats, and the healing speed was usually fast. That is why there was no advantage of suturing. Therefore, in this study, the wound was covered only with the clot at the time of hemostasis without suturing.

  1. Line 120 -…After the rats were sacrificed…”

How long after implantation were they sacrificed?

> Response: Thank you for your question. It is “3,5,7,14,28days after”. And I added the following sentences, “…Each days after the rats were sacrificed”.

  1. How many animals were used for microCT and how many for histology?

> Response: We have added the following sentences in “Statistical analysis”.

“…2.12. Statistical analysis Our experiment used 5 samples and all deta were expressed as means ± standard deviation (SD) from at least three independent experiments.”

  1. Line 137- “…The samples were next embedded in O.C.T. compound (Sakura Finetek…”

What is O.C.T.?

> Response: We have change “O.C.T” to “Optimal Cutting Temperature compound” in “Materials and Methods”.

  1. Line 144 - “ …Sections prepared from the CO3Ap group 5 days after tooth extraction…”

And the samples with AB and with the empty alveoli were not analyzed?

> Response: At first, we observed the changes of bone formation over time by HE staining, and as a result, many osteoclast-like cells were observed in the section 5 days later. Therefore, only 5 days were immune-stained and the data were shown. We have added explanation about this as follows.

“Five days after tooth extraction, HE staining revealed cells accumulating around CO3Ap (Fig. 4A). Large cells with multiple nuclei were observed. Further, cathepsin K was used to stain continuous sections, and positive cells were observed around CO3Ap (Fig. 4A). 4B) ”.

[Results]

  1. 1.AB: autogenous bone from cranium”

On the materials, is it not mentioned that AB from cranium was used?

> Response: It was our mistake. We have corrected to “autogenous bone from bovine teeth.”. Thank you for your pointing out.

  1. On Fig 2A the analysis was performed at 3, 5 and 14 days, but in the figure 2D and E the authors presents results at 3, 5, 7, 14 and 28 days. What should we accept? On materials and methods this is not mentioned!

> Response: We are sorry for the misleading description. We have modified the Fig 2A.

[Discussion]

  1. Line 234 - “…In the healing of human extraction sockets, the period up…”

What is the comparison with the rat? Why were the periods of 3, 5, 7,…days used?

> Response: Thank you for your valuable suggestions. In previous study of our laboratory, we have observed the healing process of the extraction socket over time (Atsuta et al., Biomaterials 2005, Kondo PLoS One). 2014). Therefore, this time course was adopted in this study, which observed the healing of the extraction socket. This was added in “Discussion”.

  1. Line 229 - ” … many previous studies evaluating bone formation using tibia and skull models [26].” The reference 26 does not support the sentence. The author should refer articles on studies in tibia and cranium. Like for example: Ceramics International 2018; 44 (5): 5025-5031.DOI: 10.1016/j.ceramint.2017.12.099,

> Response: Thank you for your comments. We have added the references in the text. (page 7).

[Conclusion]

The authors must replace “AB” with autogenous bone

> Response: Thank you. We have corrected it.

Line 319: “ …Although CO3Ap is an artificial bone…”

This is a synthetic material, not artificial bone. Correct this sentence.

> Response: Thank you for your pointing out the mistake. We have corrected it

[Reference]

. The authors present very old articles that should be replaced by more recent references on the same subjects.

 The authors should refer:

 : Ceramics International 2018; 44 (5): 5025-5031.DOI: 10.1016/j.ceramint.2017.12.099

> Response: Thank you for your advice. We added the some new references as reviewer suggested one. (page 7).

Reviewer 3 Report

In the manuscript “Replacement process of carbonate apatite by alveolar bone in a rat extraction socket’ authors investigated a bone graft substitute containing carbonate apatite (CO3Ap) to analyze bone replacement and the state of bone formation in vitro and in vivo compared with autogenous bone (AB).  

Overall, the experimental work is done well, and could be considered for acceptance but not in its current form. Having said that following revisions are suggested;

Comments:

  • The abstract is descriptive and qualitative.
  • Normally an abstract should state briefly the purpose of the study undertaken and meaningful conclusions based on the obtained results. Hence, this needs rewriting. I would expect brief, yet concise, the quantitative data description of the results in the abstract.
  • The given list of keywords is superficial with broader terms. More specific terms should be used. Replace accordingly.
  • The introduction is short. More literature should be added with recent and relevant literature.
  • Introduction section needs to be elaborated, in particular, the authors should highlight the novelty of this work, and illustrate the superiority of this work from previous reports, since lots of related reports have been appeared in the scientific literature. In addition, what are the gaps in this regards.
  • Authors should present the analysis of statistical significance of differences in the results section.
  • The manuscript should be carefully revised so that the results are better discussed. In my opinion, authors mainly focused on results and the Discussion section lacks scientific depth.
  • All sections should be critically discussed and compared with the previous reports. This will actually strengthen the manuscript and will highlight the significance of the work.
  • The conclusion is superficial. Herein, I would like to see the major findings and how they are addressing the left behind research gaps and covering current challenges.
  • Some sentences are long and badly worded with repetitive words. Please consider breaking longer sentences into smaller fragments for easy understanding.

Author Response

  1. The abstract is descriptive and qualitative.

Normally an abstract should state briefly the purpose of the study undertaken and meaningful conclusions based on the obtained results. Hence, this needs rewriting. I would expect brief, yet concise, the quantitative data description of the results in the abstract.

> Response: Thank you for your advice. Because there was a lot of analysis in this experiment, the detail of experimental results could not be posted. However, we have improved as much as possible. I hope you are satisfied with this change.

  1. The given list of keywords is superficial with broader terms. More specific terms should be used. Replace accordingly.

> Response: We have changed the keyword from “Carbonate apatite; Extraction socket; Bone augmentation; Autogenous bone; Bone formation” to “Carbonate apatite; Bone substitute; Autogenous bone; Osteoclast”.

  1. The introduction is short. More literature should be added with recent and relevant literature.

Introduction section needs to be elaborated, in particular, the authors should highlight the novelty of this work, and illustrate the superiority of this work from previous reports, since lots of related reports have been appeared in the scientific literature. In addition, what are the gaps in this regards

> Response: Thank you for your advice. We agree with you, and we added some references and information in “Introduction” as much as possible. If you need more, we'll get back soon.

  1. Authors should present the analysis of statistical significance of differences in the results section.

> Response: Thank you for your comment. We added descriptions of the statistically significant difference after the figure in “Results”.

  1. The manuscript should be carefully revised so that the results are better discussed. In my opinion, authors mainly focused on results and the Discussion section lacks scientific depth.

All sections should be critically discussed and compared with the previous reports. This will actually strengthen the manuscript and will highlight the significance of the work.

> Response: Thank you for your valuable feedback. We are sure that your comment let our research to increase the value. The results (especially the histological images) have been partially replaced, and information on observation findings has been added as much as possible.

  1. The conclusion is superficial. Herein, I would like to see the major findings and how they are addressing the left behind research gaps and covering current challenges.

> Response: Thank you for your comment. I tried to improve as much as possible as your advice. Please confirm these.

  1. Some sentences are long and badly worded with repetitive words. Please consider breaking longer sentences into smaller fragments for easy understanding.

> Response: Simplifying the text as much as possible, we proofread the text again. Thank you for pointing out.

Round 2

Reviewer 1 Report

The Authors have addressed all the point raised by me and other reviewers.

In my opinion, the paper is now suitable for publication on Materials Journal.

Good job.

Reviewer 2 Report

The authors did the necessary changes referred  by the reviewer. The article is suitable for publication.